# Evaluation of Barley Cultivars for Competitive Traits in Southern New South Wales

**DOI:** 10.3390/plants11030362

**Published:** 2022-01-28

**Authors:** James M. Mwendwa, William B. Brown, Paul A. Weston, Leslie A. Weston

**Affiliations:** Graham Centre for Agricultural Innovation, Charles Sturt University, Locked Bag 588, Wagga Wagga, NSW 2678, Australia; plantinteractions@csu.edu.au (W.B.B.); pweston@csu.edu.au (P.A.W.); leweston@csu.edu.au (L.A.W.)

**Keywords:** PAR light interception, weed suppressive, crop competition, canopy traits, herbicides, weed seedbank

## Abstract

The potential of multi-purpose barley (*Hordeum vulgare* L.) cultivars to suppress weeds while maintaining optimal yield and grain quality has been reported but not recently evaluated in replicated field trials performed under southern Australian field conditions. Therefore, to investigate this potential, aboveground competitive traits were assessed in nine genetically diverse commercial barley cultivars in 2015, 2016 and 2017, in two locations in the Riverina region of NSW in replicated field trials performed in the absence of pre-emergent herbicide treatment. Crop and weed establishment, early vigour, leaf area index, photosynthetically active radiation (PAR) and biomass were assessed at various crop growth stages, including early growth, vegetative, flowering, grain fill and harvest. Cultivar differences in crop and weed biomass accumulation at ~50, 100 and 150 days after planting were noted at both locations. Early barley biomass accumulation was inversely related to weed biomass in both locations and most years, suggesting strong (over 90%) potential for heritable competitive barley interference against weeds. The current study also observed a positive relationship between PAR light interception and crop biomass in all three years at both locations, suggesting that PAR light interception contributed positively to crop biomass accumulation by directly increasing photosynthesis (50–70%) and growth or indirectly influencing weed biomass accumulation (10–15%) and weed interference (50–75). Partial least squares modelling was performed with 2015 and 2016 datasets to assess the interactions between crop developmental traits and weed suppression. Cultivars exhibiting enhanced early vigour and PAR light interception were generally more weed suppressive under optimal higher soil moisture conditions. Our results indicate that the choice of barley cultivar has a significant impact on weed establishment, fecundity and seedbank dynamics. The use of competitive barley genotypes is, thus, a cost-effective strategy to reduce weed seedbank numbers over time and may reduce potential herbicide use.

## 1. Introduction

Globally, yield losses of approximately 35% are caused by weed infestation in major food crops, and loss is typically higher than those due to other pest infestations [1,2]. In addition to increased production costs [3], weeds reduce crop quality and yield. Herbicides are currently the most widely used tool for weed management in Australian grain crops. Still, over 30 key weed species have developed resistance to most herbicide sites of action and associated products [4], further limiting options for chemical control.

In Australia, herbicide resistance in both annual grasses and broadleaf weeds is currently on the rise, with resistance to multiple herbicides reported in an increasing number of cropping weeds [2,5]. The estimated cost of additional herbicide application due to resistant weeds has recently been reported to be AUD 187 million above the total cost of other integrated weed management practices [6]. The overall cost of weeds to Australian grain growers is currently estimated at AUD 3.3 billion annually, equating to AUD 146/ha for weed management and yield losses of 2.76 million tonnes of grain annually [2,6]. In contrast to chemical control options, competitive crops may be one integrated weed management tactic to consider in minimising in-crop weed infestations cost effectively [5,7]. Crop cultivars that reduce weed growth (i.e., allelopathic or competitive crops) are now frequently under consideration because they can provide a sustainable management strategy with reduced in-crop establishment costs [8].

A competitive crop can be defined [9] as one that maintains its yield in the presence of weeds (tolerant of competition) or as one that can reduce weed growth over time (by suppression of competitors) [10]. Weed suppressive ability cannot generally be attributed to a single crop growth trait but the collective effect of at least several traits [2]. In general, crop competitiveness in cereals and agronomic crops has been associated with rapid crop emergence [11], early and abundant tillering [12], high leaf area index (LAI) [13,14] and increased canopy height [15,16].

In recent years, Seavers and Wright performed research on competitive barley [14] and reported that barley competitiveness was associated with greater leaf area and height; resistance to loss of tillers under competitive pressure; and an extensive crop canopy with vigorous early establishment [2]. Thus, weed suppression in cereals has been reported to be dependent on both cereal phenology and biomass accumulation [17], and cultivars with rapid germination and vigorous growth, high biomass accumulation and efficient nutrient uptake are often the most successful [18].

Recently, there has been increased interest in selection for grain crops with improved weed suppressive ability in response to the need for sustainable weed management strategies associated with increasing environmental concerns, herbicide residue restrictions in grain export markets and the unmet needs of organic producers and smallholder farmers without access to herbicides [19] (Weston personal communication). Past research has found that the combined effects of crop competition and allelopathy determine the total weed-suppressive potential of a crop cultivar, and recent research has attempted to improve both competitive features and allelopathic potential simultaneously to achieve maximum gains in crop weed suppression [2,20,21]. In a research trial evaluating weed suppressive barley (*Hordeum vulgare* L.) cultivars, Christensen [16] found no correlation between varietal grain yields and their competitiveness with weeds. This suggests that breeding to optimise yield while maintaining competitive ability may be achievable as the traits are not linked.

Barley has been reported to be one of the most competitive cereal crops with established weeds, but competitive ability clearly varies among cultivars [22,23]. The ability to withstand competition (AWC) from winter wild oat and German-madwort in both six-row and two-row cultivars ranged from 33.7 to 78.3% and 26.7 to 69.2%, respectively. In this case, barley cultivars Ligne 640 and Pistacho (2B-1Y-1B) had high AWC from winter wild oat and German-madwort and provided high grain yields in both weedy and weed-free conditions [23] in contrast to the other cultivars in the study.

Earlier studies by Bertholdsson [24] identified competitive traits in spring barley and wheat *(Triticum aestivum* L.) and attempted to determine their importance in subsequent weed suppression. Weed biomass 1–2 weeks before sheath emergence was used as a measure of genotypic differences in competitiveness against weeds. Early crop biomass explained 24–57% of the observed genotypic variance across four years of experimentation; allelopathic activity explained 7–58%; and, combined, both traits explained 44–69% of the observed genotypic variance for weed suppression. Corresponding estimates were much lower in wheat: 14–21% for early biomass, 0–21% for allelopathic activity and 27–37% when combined [24]. Recent studies by Mwendwa et al. [2,25] showed that early vigour and early canopy closure help suppress weed establishment and growth in winter wheat, and wheat seedlings in the field produced higher concentrations of root exuded allelochemicals earlier in the season. These findings suggest that combined effects of both competitive crop ability and allelopathic activity contribute to weed suppression in cereals. We also understand that significant differences in the weed suppressive abilities of cereal crops have been observed, with oats typically being the most suppressive, followed by barley and then wheat [14].

Didon [11] reported that barley cultivars with strong to medium competitive ability against weeds shortened the time of emergence after planting in the presence of annual broadleaf weeds, in contrast to the emergence in the least competitive cultivar. Spring barley cultivars with strong competitive ability against weeds also exhibited early stem extension as a response to weed competition. Morphological traits, including length of the two first internodes, long main shoot in the tillering stage and a slight leaf angle, were also suggested to be essential traits in competition for light [2,11].

Based on the findings of previous studies, the aims of the present experiments were (i) to identify the variation in the early competitive response to weed competition of selected genetically diverse Australian commercial barley cultivars at two locations across New South Wales (NSW); (ii) to evaluate crop canopy architecture traits contributing to crop competition at different growth stages; and (iii) to investigate whether differences between the barley cultivars in shoot morphology, growth and development and competitive response were correlated with differences in competitive ability against weeds.

## 2. Materials and Methods

### 2.1. Site Description and Experimental Design

From 2015 to 2017, replicated barley field trials were sown at two NSW locations with well-drained red kandosol soils in moderate to low rainfall zones in Wagga Wagga (572 mm) and Condobolin (449 mm), respectively [2]. Plots were seeded similarly with six replications arranged in a randomised complete block design. In all years, nine barley cultivars commercially grown in Australia were selected for evaluation, representing feed, malt and grazing types, with some serving as dual or triple purposes. In addition, a cultivar of winter cereal rye (*Secale cereale* L.) was used as a positive for weed suppressive control (Table 1).

At Wagga Wagga, field trials were conducted on fine red clay loam sodosols, surface pH 6.4, previously planted commercially for the production of cereals, canola and/or lucerne (*Medicago sativa* L.). At Condobolin, soils were predominantly red gradational and red-brown earth sodosols with surface pH 7.0 and were previously rotated among canola, cereals and pasture legume crops. Both soils exhibited low inherent fertility and organic matter content and were maintained using standard commercial practices to reduce weed populations [26,27].

### 2.2. Crop Establishment

Crops were sown with seed generated in Wagga Wagga NSW during the previous season using standard practices for pest control and fertiliser application. This potentially eliminated significant cultivar variation due to seed variability resulting from production at different sites [2,28,29]. At Condobolin, the crop was sown on 15th, 17th and 11th May at 33 cm spacing, typical for drier soils, while at Wagga Wagga, the crop was sown on 22nd, 14th and 22nd May at 25 cm spacing in 2015, 2016 and 2017, respectively, due to soil moisture differences encountered among the locations. Cultivars were established at equal plant density (target population of 120 plants/m^2^) in each trial by sowing seed lots adjusted for seed weight per cultivar and germination rate. A total of 200 seeds of each cultivar were weighed, and the weight was multiplied by five to determine the seed weight/1000. Each cultivar’s calculation was performed to determine the sowing rate for achieving the target plant population [2,29], as shown in the Equation (1) below.
Sowing rate (kg/ha) = (Seed weight/1000 (g) × Proposed target plant population (plants/m^2^) × 100)/(Expected Establishment % × Actual germination percentage)(1)

Plots were planted using a calibrated disc cone seeder (Kimseed Australia Pty Ltd., Wangara, Australia) with a 22 and 30 cm row spacing and 25 and 33 cm in-row spacing between plants, respectively, at Wagga Wagga and Condobolin. Seeds were sown together (in-row) with granulated fertiliser; diammonium phosphate (DAP) (analysis 17.1% N, 20.0% P; Incitec Pivot Fertilisers, Melbourne, Australia) was applied at 70 kg/ha. DAP was treated with 400 mL/ha Flutriafol (Intake^®^Hiload Gold 200 g/ha Flutriafol, Nufarm Australia, Melbourne, Australia) [2]. Before sowing, all the established weeds were controlled with glyphosate (Weedmaster^®^DST^®^ 470 g/L Glyphosate, Nufarm Australia, Melbourne, Australia) at 960 g/ha. Individual plots measured 2 × 12 m and were trimmed to 2 × 10 m after crop establishment [2].

### 2.3. Crop Assessments and Data Collection

Data collection was performed at each location with evaluation periods based upon critical plant developmental stages of crop establishment, including stem elongation, flowering and maturity. Data were collected each year on various crop phenological characteristics, including leaf size, early vigour and crop canopy at approximately 50, 100 and 150 DAE [2]. A light ceptometer (AccuPAR LP-80 Ceptometer, Decagon Devices^®^, Pullman, WA, USA) was used to measure PAR (photosynthetically active radiation—µmols m^2^s^−1^) both above and below the crop canopy, light interception (%) and leaf area index (m^2^) [2]. Leaf area index and light interception (%) below the crop canopy were calculated from above and below canopy PAR readings. All measurements were made between 11 h and 14 h (solar noon at 13 h) [2].

NDVI (normalised difference vegetative index) readings (GreenSeeker^®^ 505 handheld sensor and Trimble^®^ Recon PDA, NTech Industries Inc., Sunnyvale, CA, USA) were obtained (at 1 to 1.5 m above ground) to monitor canopy closure and estimate crop biomass production [2]. NDVI is preferable for global vegetation monitoring since it compensates for changes in lighting conditions, surface slope, exposure and other external factors. Low NDVI values indicate moisture-stressed or sparse vegetation, and higher values indicate healthy and dense green foliage. NDVI was calculated following the formula in Equation (2) below [2]:NDVI = NIR − RED/NIR + RED(2)
where NIR is a reflection in the near-infrared spectrum, and RED is a reflection in the red range of the spectrum.

Other assessments included crop biomass and weed biomass in-crop, visual vigour ratings (0 = poor, no stand; 5 = crop with more than 50% open canopy space; 10 = high vigour, closed canopy) based on crop growth and biomass accumulation over time, specifically ground cover resulting in canopy closure (a 10 would be recorded for an extremely vigorous crop with closed canopy in contrast to a less vigorous crop with open canopy allowing light to reach the soil surface) [2]. Post-harvest weed suppression visual ratings were also performed (0 = weeds absent; 5 = crop and weeds at 50:50 ratio; 10 = weeds dominate, no crop) [7,25]. Time units in all results are expressed as days after crop emergence (DAE) [2].

Aboveground crop biomass, total weed numbers (overall and per significant individual species) and weed biomass were measured in two 50 × 50 cm quadrats per plot in all 6 replicates generally at four critical growth stages: early growth (30–40 days following establishment), vegetative growth (60–70 days following establishment), flowering and crop harvest [2]. Biomass was obtained by cutting plant material at the soil surface and weighing it after drying at 40 °C for 5 days in a forced-air oven. Weed counts were monitored in-crop and after harvest by counting identified weeds in two 50 × 50 cm quadrats per plot [2]. To increase the robustness of the data collected to estimate weed numbers and biomass, an appropriate number of sub-samples (6 × 2 replicates = 12 subsamples) in each cultivar treatment was obtained. Sub-sampling in six replications has proven useful in past experimentation at both locations, given the inherent variability of natural weed populations in field conditions [2]. Grain harvest was performed at crop maturity before 15 December in each year and location, using a small plot harvester (plot harvest area = 18 m^2^). Yield was measured as harvested cereal grain in metric tons per hectare (t/ha) [2].

### 2.4. Data Analysis

Trial design, randomisation and data analyses were performed using Agricultural Research Manager (ARM) version 9.0 (Gylling Data Management Inc. 2014) [2]. GenStat was used to perform all statistical analyses of selected datasets [30] for ANOVA and MANOVA REML (residual/restricted maximum likelihood) or regression model with means separated using LSD (0.05 confidence level) [2]. The efficacy of each cultivar in terms of weed suppression was calculated using Equation (3) based on the least weed suppressive cultivar in the same year and location [2] (data presented in Table 1):C_e_W_s_ = (C_l_W_b_ − C_t_W_b_) ÷ C_l_W_b_ × 100(3)
where C_e_W_s_ is cultivar effectiveness in weed suppression, C_l_W_b_ is the value of weed dry biomass for the cultivar with the least weed biomass in the year and location and C_t_W_b_ is the value of weed dry biomass for the cultivar.

Data were transformed using the square-root ((x + c)^0.5^) function to correct for homogeneity and normality before analysis as required. Partial least square (PLS) regression analysis was performed using PLS-R (XLSTAT software, version 9.0 by Addinsoft, Paris, France).

Data produced in this study were used to investigate multiple interactions both at the crop and weed environmental levels. To analyse this complex dataset and due to its display of multicollinearity (e.g., PAR, LAI and height), a model was generated that utilised partial least squares (PLS) regression or (PLS-R) as the statistical model [2,31]. PLS regression is a technique that reduces the predictors to a smaller set of uncorrelated components and performs least squares regression on these components instead of on the original data. This analysis is particularly useful when predictors are highly collinear or when there are more predictors than observations [2,31]. The interrelatedness of plant characteristics associated with canopy structure and weed competitiveness in grain crops (i.e., plant height, early canopy closure, LAI, vertical leaf orientation, rapid biomass accumulation at the early crop growth stage, high shoot dry matter, large root biomass and root volume [32]) makes PLS regression analysis particularly well-suited to characterise relationships between crop plant characteristics and weed suppression [2].

Statistical analysis modelling of the data was performed by linear and partial least squares (PLS) regression for randomised experiments with four replicates using XLSTAT (Addinsoft, New York, NY, USA). Dry weed biomass was used as the dependent variable (or variable to model). In contrast, the quantitative explanatory variables included crop biomass, PAR light interception expressed as a percentage of PAR light intercepted by crop canopy at the sampling time, leaf area index, visual vigour ratings and NDVI, taking into account the time of sampling and crop growth stage. PLS results are presented on the variable importance in model projection (VIP) charts (one bar chart per component); a border line is plotted to identify VIPs that are greater than 0.8 and above. These thresholds allow identification of the variables that are moderately (0.8 < VIP < 1) or highly influential (VIP > 1) [2,31,32]. The VIP score first published by Wold and others in 1993 measures the explicative power of predictor variables with respect to the response variable based on PLS-R. The VIP score of variable j is calculated by Equation (4) below [2]:
(4)$$\mathrm{VIPj}=\sqrt{\frac{\sum_{\begin{array}{c} a=1 \end{array}}^{h}R^{2}\;\left(y,ta\right){\left(Waj/\backslash \backslash wa\backslash \backslash \right)}^{2}}{\left(\frac{1}{p}\right)\sum_{\begin{array}{c} a=1 \end{array}}^{h}R^{2}\left(y,\;ta\right)}}$$
where *Waj* is weight of the *j*th predictor variable in component a, and *R*^2^(*y*,*ta*) is a fraction of variance in *y* explained by the component *a*. The variable with a higher VIP score indicates that it is more relevant to predict the response variable [2,31,32].

## 3. Results

### 3.1. Monthly Rainfall at Both Locations

Figure 1 and Figure 2 report the monthly rainfall received during the growing season in Wagga Wagga and Condobolin NSW from 2015 to 2017. Barley cultivars demonstrated variable performance based upon the below average (2017), average (2015) and above-average (2016) rainfall received across years and locations. The in-crop rainfalls received in Condobolin were 331 (2015), 539 (2016) and 226 (2017) with a long-term average of 288 mm, while in Wagga Wagga, the variation was more significant with 425 (2015), 655 (2016) and 266 (2017) received with a long-term average of 372 mm. However, greater weed pressure was observed based on weed count and biomass assessments at both locations (Figure 3C by 60% at Condobolin and D by 1000% at Wagga Wagga) due to the above-average in-crop rainfall in 2016. In contrast, rainfall in 2017 was lower than the long-term average by 21.6 and 28.5%, respectively, at Condobolin and Wagga Wagga. The exceptionally dry conditions in 2017 resulted in reduced crop yield (up to 50% less at both sites) and poor weed establishment (5 to 10% groundcover or less at both sites); hence, no weed biomass was collected at the experimental sites in 2017 (Figure 3E,F).

### 3.2. Early Competitive Ability of Barley to Suppress Weeds Based on Biomass Accumulation

Figure 3 is a multi-panel graph and presents crop and weed biomass taken after crop emergence (at approximately 50, 100 and 150 days). At Condobolin in 2015 (Figure 3A,C,E), there were significant cultivar differences in crop biomass at 50 (*p* < 0.05) and 150 DAE (*p* < 0.01). At both sampling dates, cultivars Compass, Commander and Litmus produced up to 20–30% higher crop biomass and subsequently improved weed suppression with reductions in up to 45% at 100 DAE. However, major differences in weed suppression at 100 DAE were not observed in 2016.

In 2015, at Wagga Wagga (Figure 3B,D,F), early crop biomass differed between cultivars at both 50 (*p* < 0.001) and 100 DAE (*p* < 0.001), with Navigator, Compass and Litmus producing up to 10–20% higher average crop biomass and similar reductions in weed biomass (Figure 3). However, there were no differences in crop biomass at 150 DAE. In 2016, above-average rainfall was noted in both locations, and no significant differences between cultivars were noted with respect to crop biomass as soil moisture was not limiting barley growth and canopy development. Subsequently, limited differences in weed interference were noted in 2016 among cultivars at 100 DAE.

### 3.3. Barley Canopy Architecture Traits Contributing to Weed Suppression at Different Growth Stages

Figure 4 presents a comparative regression analysis between PAR light interception and crop biomass at both locations over each year. A strongly positive relationship was noted in all years (*p* < 0.001) at both locations, with the only exception being Wagga Wagga in 2016 where this relationship was weak (Figure 4). Here, the relationship was impacted by the timing of PAR reading and crop biomass collection, year and location. At Condobolin, the coefficient of determination (*R*^2^) was 0.75 (*p* < 0.001, Figure 4A), 0.55 (*p* < 0.001, Figure 4C) and 0.25 (*p* < 0.01, Figure 4E) in 2015, 2016 and 2017, respectively. Similarly, at Wagga Wagga, the coefficient of determination (*R*^2^) was 0.50 (*p* < 0.001, Figure 4B) in 2015 and 0.67 (*p* < 0.001, Figure 4F) in 2017. However, *R*^2^ was markedly lower in 2016 at 0.06 in the year with high soil moisture, showing limited association and suggesting moisture use efficiency may effectively drive canopy formation and biomass accumulation in certain cultivars in dry years (Figure 4D).

### 3.4. Correlation between Barley Cultivar Selected Canopy Traits and Competitive Ability against Weeds

Figure 5 and Figure 6 report the PLS regression results from the predictive linear model performed to demonstrate weed suppression in commercial barley cultivars based on weed dry biomass in each cultivar as the response variable and selected aboveground crop canopy traits data (crop biomass, PAR light interception, leaf area index, visual vigour ratings, plant height and NDVI) as the predictive variables in 2015 and 2016. Unfortunately, modelling was not performed for 2017 data as weed biomass was not collected due to lack of weed growth associated with prolonged drought.

The PLS regression model prediction differed in 2015 in accordance with crop canopy traits and their collective impact on weed suppression resulting in lower weed biomass at both locations (*p* < 0.001, Figure 5). In Condobolin, the model fit was R^2^ = 0.14 with early growth vigour, NDVI, PAR light interception, leaf area index, crop biomass, crop height and plant counts being inversely correlated with weed biomass. Similarly, at Wagga Wagga PAR light interception, leaf area index, crop biomass, crop height and plant counts were strongly inversely correlated with weed biomass. However, in Wagga Wagga the plant count correlation was more robust than in Condobolin, with R^2^ being 0.10.

In 2016, at both locations, plant count, leaf area index PAR light interception and crop biomass were strongly and inversely related to weed biomass (Figure 6) with a model fit coefficient (r^2^) of 0.25 and 0.16 for Condobolin and Wagga Wagga, respectively. If the crop was able to achieve a closed canopy before heading, weed suppression was generally significantly improved by 25%. The cultivars that established well typically exhibited early growth vigour and reduced PAR light interception and were more weed suppressive in 2015 and sometimes in 2016, despite the very high soil moisture levels in 2016.

### 3.5. Barley Cultivar Grain Yield and Tolerance against Weeds

Grain yield varied with cultivar and location (*p* < 0.001, Table 2). At Condobolin, there were significant differences in yield between cultivar and years, while at Wagga Wagga, there were no major cultivar differences in yield in 2016. Generally, the highest yields (Table 2) were recorded at the Wagga location in 2015 (3.4 t ha^−^^1^ more) and 2016 (1.6 t ha^−1^ more), likely due to improved soil moisture and environmental conditions compared to Condobolin, which received less rainfall. At Condobolin, Compass, Hindmarsh, Litmus, Navigator and Urambie performed well with respect to yielding ability in both years, while at Wagga Wagga, Hindmarsh was the most consistent performer in terms of yield, followed closely by La Trobe and Compass. Commander, Navigator and Westminister performed well in terms of yield in 2016 under higher soil moisture levels. Interestingly, yield was most impacted by soil moisture availability and was up to 4-times greater in Wagga Wagga and Condobolin in 2016 when soil moisture was not limiting in contrast to 2015 and 2017 conditions.

## 4. Discussion

### 4.1. Early Competitive Ability of Barley to Suppress Weeds Based on Biomass Accumulation

In the current study, apparent differences were identified in biomass accumulation between winter barley cultivars, indicative of the strong impact of genetics along with location and seasonal differences experienced over 2015, 2016 and 2017. Early cultivar biomass accumulation was strongly and inversely correlated with weed biomass in both locations and most years, suggesting potential for heritable competitive ability against weeds [2]. Christensen [16] reported similar findings while studying faster-growing cultivars of spring barley that proved more weed suppressive. In addition, Coleman and Mwendwa et al. [2,33] reported recently that more rapid leaf area development and early crop vigour improved winter wheat’s ability to compete with weeds.

Didon [11] also reported that the six most competitive six-row barley cultivars grown in competition with the annual broadleaf weeds reduced the fresh weight of the weeds by 61–75% compared with the least competitive six-row cultivars. In both barley and wheat, multiple regression analysis revealed that early crop biomass and potential allelopathic activity were the only parameters that significantly contributed to competitiveness. In barley, early crop biomass explained 24–57% of the observed genotypic variance across four years [24]. In addition, differences in cultivar competitiveness were cultivar-specific and not generic [34].

In the current study, we noted that when soil moisture was not limiting (e.g., in 2016) in both locations, there were no highly significant differences between barley cultivars in either crop biomass or weed interference under lower inputs experienced in southern Australian production systems. In contrast, in an average year such as 2015, cultivar differences in weed interference and biomass accumulation were highly significant, with cultivars such as Litmus, Compass and Urambie supporting from 20 to 50% less weed biomass than other cultivars by 100 DAE. At this time, canopy closure had occurred in these cultivars, and weed growth was further limited by low light conditions and in-crop competition for resources. Under Australian conditions encountered in the Riverina region of NSW, soil moisture availability may play a critical role in differential barley competitive ability against weeds. Mwendwa et al. also reported similar observations in wheat [2] and canola [27]. In 2015 and 2017, weed pressure was significantly reduced in contrast to 2016; this was likely associated with limited soil moisture availability or drought experienced in both locations. However, our results have demonstrated that more competitive cereal cultivars contributed to significant weed biomass reductions and with respect to annual grass weeds such as *Lolium rigidum*, which particularly reduced seed set in 2015–2017, as evidenced by soil seedbank experiments conducted by replicated soil sampling of competitive rotational crop selections over time. Under conditions of non-limited soil moisture, cereal cultivars established closed canopies more quickly but weed pressures were higher, and limited differences were observed among barley cultivars that are typically quite competitive with weeds compared to wheat, canola or triticale. Therefore, we recommend further investigation of the impact of soil moisture availability on competitive interference of barley with annual weeds.

### 4.2. Barley Canopy Architecture Traits Contributing to Weed Suppression at Different Growth Stages

This study demonstrated cultivar differences in crop growth and visual vigour rating, plant density, PAR light interception (%), leaf area index, NDVI and plant height with cultivar, year and location interactions. Competitive crop canopy traits have been studied to determine their relative contribution to weed suppression in conventional agricultural systems where fertilizer and water supply are not restricting plant growth. Highly competitive barley cultivars typically can access light, nutrients and water resources efficiently in a limited spatial area, thus effectively suppressing the growth and reproduction of neighbouring weed species [35,36]. Plant traits associated with canopy structure and weed competitiveness in cereal crops include plant height, early canopy closure, LAI, vertical leaf orientation, rapid biomass accumulation at the early crop growth stage, high shoot dry matter, large root biomass and root volume [2,37].

The current study observed a highly positive relationship between PAR light interception and crop biomass in all three years at both locations. This suggests that PAR light interception may contribute positively to crop biomass accumulation perhaps by increasing photosynthesis and growth, directly and/or indirectly influencing weed biomass accumulation and weed interference. Didon and Hansson [38] also demonstrated that the most weed-suppressive barley cultivars intercepted the greatest PAR light. This finding suggests that assessment of PAR light interception may present a simple and useful method to rank cultivar suppressive ability for future trait identification for pre-breeding analysis. Additional studies should be performed to verify the validity of this relationship at multiple locations.

### 4.3. Modelling—Correlation between Barley Canopy Traits and Competitive Ability against Weeds

The interrelatedness of plant characteristics associated with canopy structure and weed competitiveness in grain crops (i.e., plant height, early canopy closure, LAI, vertical leaf orientation, rapid biomass accumulation at the early crop growth stage, high shoot dry matter, large root biomass and root volume [2,32]) renders PLS regression analysis particularly well-suited to characterise relationships between crop plant characteristics and weed suppression [37].

A model was generated in 2015 and 2016 at each location to determine which crop traits influenced weed biomass accumulation. In particular, we noted that early vigour and plant establishments at 100DAE were strongly associated with weed biomass, as was PAR light interception at the soil and canopy surface. The model outcomes suggest that early crop biomass is in itself not a standalone trait but a combination of other cultivar traits such as leaf area index, crop height, tillering and early canopy closure may likely impact canopy formation and light interception and the resulting ability to suppress weed growth. The cultivars that established well also exhibited early growth vigour and reduced PAR light interception and were typically more weed suppressive in 2015 and sometimes in 2016 despite increased soil moisture levels and weed burden in 2016. Paynter and Hills [34] reported that increasing barley plant density increased grain yield and reduced both rigid ryegrass dry matter and tiller number. Interestingly, barley density had a larger impact on rigid ryegrass productivity than the choice of cultivar.

A better understanding of the interaction between key barley developmental characteristics will assist breeders in selection of more weed suppressive cereal crop genotypes. The findings of our studies suggest that early vigour (the ability of the crop to shade the soil by canopy architectural traits) expressed before crop maturity (at 50 to 100 days after seeding) clearly impacts weed biomass and eventual propagule formation [2,39], and selection for such traits can result in enhanced weed suppression with respect to currently available commercial cultivars. As new barley cultivars are frequently released, additional screening of novel genotypes will be required more globally and regionally to predict their associated weed suppressive activity on a regular basis.

### 4.4. Barley Cultivar Grain Yield and Tolerance against Weeds

Crop competitive ability can be separated into two components: (1) the competitive effect or weed-suppressive ability, i.e., the capacity of the crop to reduce weed growth and reproductive success through interference; and (2) the competitive response or weed tolerance, i.e., the ability of the crop to yield despite the presence of weeds [40,41].

In the current study, we have observed and demonstrated that Hindmarsh and Compass were generally the best performing cultivar at both locations in terms of total yield produced. However, Hindmarsh was also among the least weed suppressive, allowing weeds to establish and set seed at a potentially greater rate than other more competitive cultivars such as Compass. Results generated at both trial locations over three seasons suggest that growers could suppress weeds more effectively while maintaining yielding potential by judicious cultivar choice.

## 5. Conclusions

The current study demonstrated that (1) early cultivar biomass accumulation was strongly and inversely correlated with weed biomass in both locations and most years. (2) When soil moisture was not limiting, they were limited to no significant differences between cultivars in crop biomass or weed interference, but in an average year, cultivar differences in weed interference and biomass accumulation were highly significant between cultivars and locations. (3) A strong positive relationship between PAR light interception and crop biomass was shown, suggesting that PAR light interception may contribute positively to crop biomass accumulation directly and/or indirectly influencing weed biomass accumulation and weed interference. (4) Early vigour and plant establishments at 100DAE were positively associated with weed biomass reduction, as was PAR light interception at the soil and canopy surface, as supported by PLS modelling (5). Compass was the most weed competitive and consistently high yielding cultivar; however, Hindmarsh performed the best in terms of yield both locations, despite its limited weed suppressive potential.

The use of competitive barley cultivars presents a potentially cost-effective strategy to reduce weed seedbank numbers over time. However, the choice of cultivar is likely to be location-dependent, suggesting that regional studies are required with newly released cultivars to select for optimal weed management choices. In addition, crop biomass accumulation and PAR light interception were critical in predicting barley weed suppression. It is likely more useful for producers to consistently consider a highly competitive seeding productive cultivar such as Compass, which produces high grain yields and allows fewer weed propagules to be established in the seedbank, resulting in potentials for reduced herbicide usage. We recommend that future barley cultivar selections consider the incorporation of weed suppressive traits, particularly since producers are highly supportive of sustainable cost-effective weed management strategies that do not require new technological interventions on-farm.

## Figures and Tables

**Figure 1 plants-11-00362-f001:**
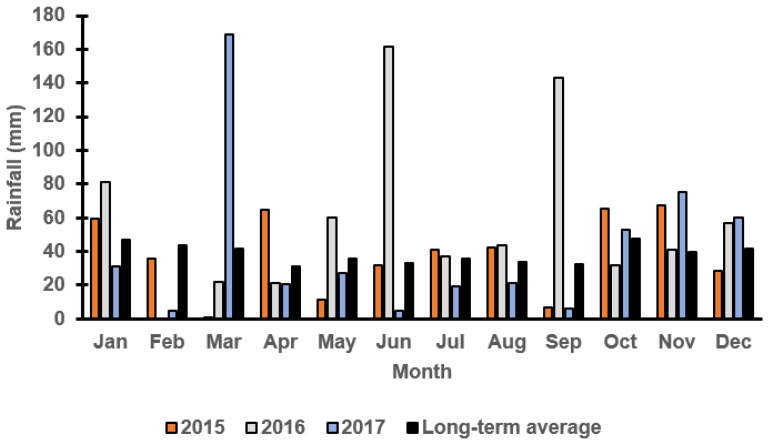
Monthly rainfall (mm) at the Condobolin field trial site in 2015, 2016 and 2017 as reported by the Australian Bureau of Meteorology. The long-term average is calculated over the last 100 years.

**Figure 2 plants-11-00362-f002:**
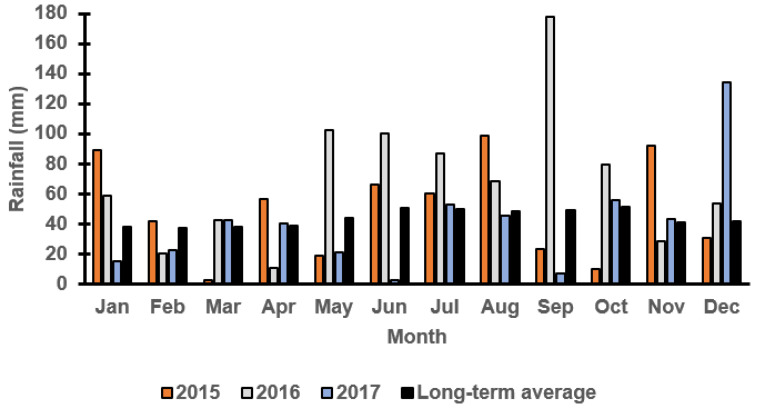
Monthly rainfall (mm) at the Wagga Wagga field trial site in 2015, 2016 and 2017 from the Australian Bureau of Meteorology. The long-term average is calculated over the last 100 years.

**Figure 3 plants-11-00362-f003:**
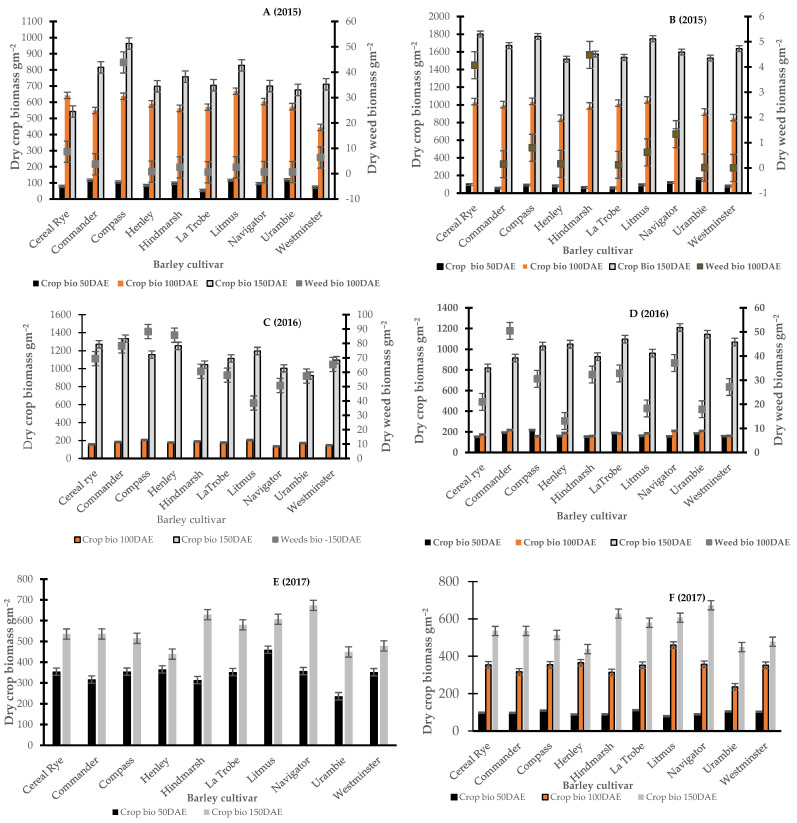
Crop and weed biomass taken at different growth stages indicted by DAE (days after emergence) at Condobolin (**A**—At 50 DAE *p* < 0.5, LSD = 42.5, 100 DAE NS, 150 DAE *p* < 0.001 LSD = 172.6; **C**—At 100 DAE NS, at 150 DAE NS; **E**—At 50DAE *p* < 0.001 LSD = 73.36, 150DAE NS) and Wagga Wagga (**B**—At 50 DAE *p* < 0.001 LSD = 36.36, 100 DAE *p* < 0.001 LSD = 127.5, **D**—50 DAE NS, 100 DAE NS, 150 DAE NS; **F**—50 DAE) in 2015, 2016 and 2017. No weed biomass was collected in 2017 due to exceptionally dry conditions experienced, which limited weed establishment and growth.

**Figure 4 plants-11-00362-f004:**
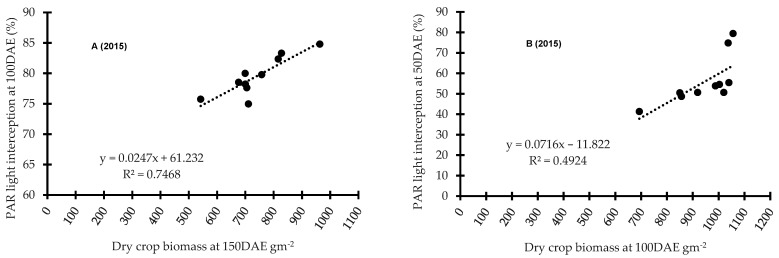
The comparison between photosynthetically active radiation (PAR) light interception and crop biomass at Condobolin (**A**,**C**,**E**) and Wagga Wagga (**B**,**D**,**F**) in 2015, 2016 and 2017 at approximately 50 to 100 DAE for PAR light interception and 50 to 150 DAE for oven dried crop biomass respectively.

**Figure 5 plants-11-00362-f005:**
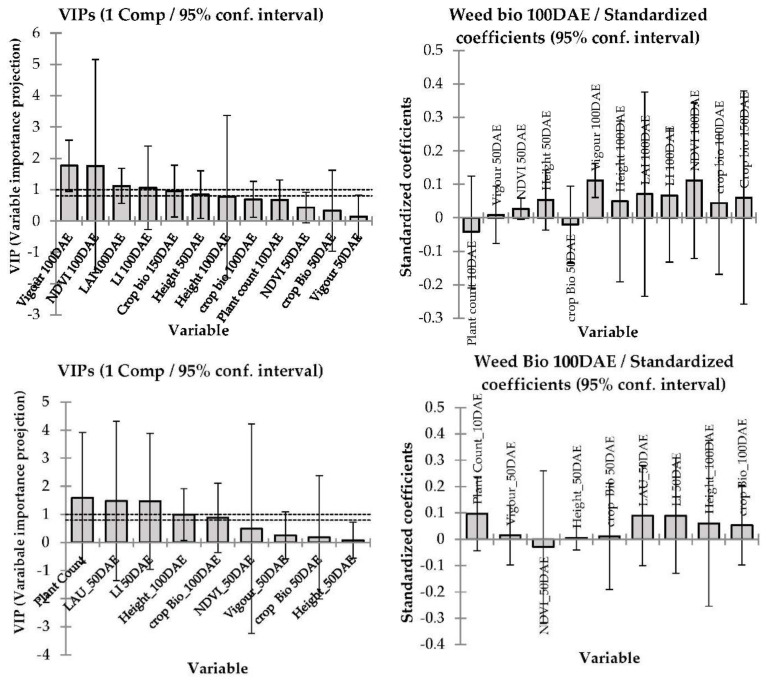
The variable importance projections and corresponding standard coefficients of different barley canopy traits determine the weed suppression in 2015 at Condobolin (above panel) and Wagga Wagga (below). The model goodness of fit relationship R^2^ = 0.14 at Condobolin and 0.10 at Wagga Wagga, respectively.

**Figure 6 plants-11-00362-f006:**
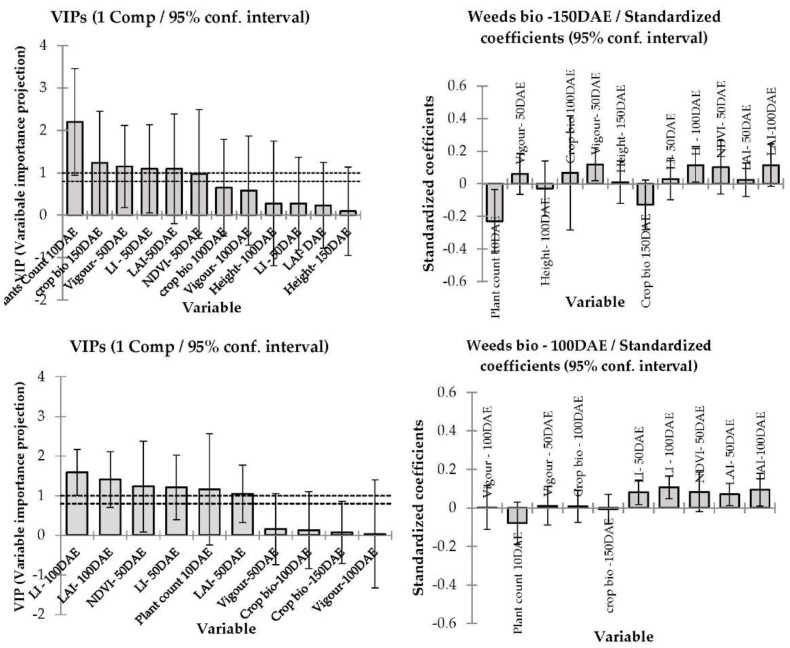
The variable importance projections and corresponding standard coefficients of different barley canopy traits determine weed suppression in 2016 at Condobolin (above panel) and Wagga Wagga (below). The model goodness of fit relationship Rr^2^ = 0.25 at Condobolin and 0.16 at Wagga Wagga, respectively.

**Table 1 plants-11-00362-t001:** Barley cultivars plus cereal rye evaluated in field trials in Wagga Wagga and Condobolin, NSW conducted from 2015 to 2017.

	Wagga Wagga	Condobolin	Description
1	Commander	Commander	Malt
2	Compass	Compass	Malt and Grazing
3	Henley	Henley	Malt
4	Hindmarsh	Hindmarsh	Feed/malt
5	La Trobe	La Trobe	Malt
6	Litmus	Litmus	Malt
7	Navigator	Navigator	Malt
8	Urambie	Urambie	Feed and Grazing
9	Westminster	Westminster	Malt
10	Grazer (Cereal rye)	Grazer (Cereal rye)	Dual purpose (control)

**Table 2 plants-11-00362-t002:** The average crop yield per cultivar in t ha^−1^ in 2015, 2016 and 2017 at Condobolin and Wagga Wagga.

Location	Condobolin Yield t ha^−1^			Wagga Wagga Yield t ha^−1^		
Cultivar/Year	2015	2016	2017	2015	2016	2017
Cereal Rye	0.4	2.2	0.9	2.7	5.7	1.3
Commander	1.3	4.3	1.4	4.0	6.3	1.1
Compass	1.2	4.6	1.8	4.5	5.4	1.4
Henley	1.3	4.5	1.5	4.3	5.6	1.1
Hindmarsh	1.4	3.8	1.5	5.1	5.7	1.5
La Trobe	1.3	4.2	1.6	4.9	5.5	1.4
Litmus	1.7	4.3	1.4	4.3	5.8	1.4
Navigator	1.1	4.7	1.6	4.4	6.1	0.8
Urambie	1.3	3.9	1.2	3.8	5.3	1.2
Westminster	1.1	3.9	1.3	3.8	6.1	1.2
LSD_0.05_	0.47	0.70	0.28	0.50	1.70	0.13
CV %	7.4	12.1	16.8	9.4	20.6	9.13
*p*-value	>0.01	>0.01	>0.01	>0.01	NS	>0.01

## Data Availability

The data presented in this study are summarised in this manuscript and raw data is held at Charles Sturt University and the Grains Research and Development Corporation.

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
