# Peer review of "Evaluation of Barley Cultivars for Competitive Traits in Southern New South Wales"

_plants, 2022, doi:10.3390/plants11030362_

Round 1
Reviewer 1 Report
Please reorganize the manuscript to have 1) introduction, 2) materials and methods, 3) results, 4) discussion with references and conclusions. Add aims or research questions in the 1 or 2 chapter.
Detailed comments:
1. Please write the research method clearly in the abstract.
2. Please write key findings in the abstract clearly and briefly.
3. Please address all stated aims of the study in the abstract (findings), results and conclusions.
4. Please do not use references in the conclusions section and write more conclusions covering all aims and all real-life, commercial and policy ect implications. Please address sustainability and all environmental impacts (e.g. climate, waters ect) as much as you can (compare to other options if possible). Add suggestions for future research.
Author Response
Reviewer 1
Thank you for your time and effort to read through our MS. Thank you also for your comments and suggestions on how we can improve the MS.
Please reorganize the manuscript to have 1) introduction, 2) materials and methods, 3) results, 4) discussion with references and conclusions. Add aims or research questions in the 1 or 2 chapter.
We have reorganised the manuscript as suggested, i.e., 1) Introduction, 2) Materials and Methods, 3) Results, 4) Discussion, 5) Conclusions and 6) References
Detailed comments:
- Please write the research method clearly in the abstract.
We have revised and rewritten the research method in the abstract lines 10-15
- Please write key findings in the abstract clearly and briefly.
We have included the key findings in the abstract with more clarity than before; please see Line 15-26
- Please address all stated aims of the study in the abstract (findings), results and conclusions.
We have addressed the stated aims of the study in the abstract by providing the results for each aim. Under results and discussion, we have provided guidance to the readers by including the aims in the subheadings. However, in conclusion, we have itemised the findings under each aim and provided a general conclusion on the application.
- Please do not use references in the conclusions section and write more conclusions covering all aims and all real-life, commercial and policy etc implications. Please address sustainability and all environmental impacts (e.g., climate, waters etc) as much as you can (compare to other options if possible). Add suggestions for future research.
Thank you for your suggestions for the conclusion. We have incorporated all of them in the current conclusion – See lines 465--475
Reviewer 2 Report
- The abstract must include information about the authors' significant findings as well as other noteworthy findings.
- In the introduction section, even though the authors provide an extensive literature review the section of the Introduction, they should enhance and reconstruct it. They should provide the scope of the paper clearly.
- In general, there is a repetition of information that might have been omitted.
- Abstract and Conclusion must be rewritten. Only 1 sentence in the abstract section represents the results, while the rest of the section deals with the problem statement.
- The conclusion is a repetition of the problem statement, which does not make a sense.
- Statistical analysis should include details on data distribution and related pre-requisites of ANOVA and MANOVA.
- Please revise your manuscript in the light of these comments in addition to the concerns raised by reviewers.
- Examine the ligands in the figure; they have been scribbled randomly.
- The discussion of important and related works should contain a larger quantity of information and references.
- For the discussion section, I suggest simplifying the style and make more readable the text, without long sentences.
Author Response
Reviewer 2
Thank you for your time and effort to read through our manuscript. Thank you also for your comments and suggestions on how we can improve the manuscript. Much appreciated
- The abstract must include information about the authors' significant findings as well as other noteworthy findings.
Thank you for your suggestion. We have included the significant findings in the abstract see line 15-26
- In the introduction section, even though the authors provide an extensive literature review the section of the Introduction, they should enhance and reconstruct it. They should provide the scope of the paper clearly.
We have addressed both worldwide and local problems in the first two paragraphs of the introduction. Then focused on competitive crops as an alternative to costly herbicides. The last paragraph in the introduction clearly states the aims of the study – see lines 107-113
- In general, there is a repetition of information that might have been omitted.
Due to your suggestion, we identified some repetitions in the abstract and other parts of the manuscript, and we have addressed them. Thank you
- Abstract and Conclusion must be rewritten. Only 1 sentence in the abstract section represents the results, while the rest of the section deals with the problem statement.
We have rewritten the abstract and conclusion and included the main findings in relation to the aims of the study. For abstract please see line 15-26, and for conclusion, please see lines 452- 473 for more details
- The conclusion is a repetition of the problem statement, which does not make a sense.
We have addressed the issue of repetition by deleting the repeated words and sentences. We hope it makes sense now.
- Statistical analysis should include details on data distribution and related pre-requisites of ANOVA and MANOVA.
Thank you for your suggestion. However, we have given in detail the statical analysis methods which we used. Please see lines 212-214. We also did data transformations where the data failed the normality test (see lines 224-225). Otherwise, what is reported in the Materials and Methods is what we did
- Please revise your manuscript in the light of these comments in addition to the concerns raised by reviewers.
We have done our best to revise the manuscript based on the reviewer comments and suggestions. These comments and suggestions are much appreciated.
- Examine the ligands in the figure; they have been scribbled randomly.
I am very sorry we did not find these ligands throughout all the figures.
- The discussion of important and related works should contain a larger quantity of information and references.
Thank you for your suggestion. I agree it is essential to include as many references in the discussion as you can get. However, it can be very limiting if there is not a lot of previous research that has been done on a similar study. Therefore at this stage, we used as many references as we could find related to our study.
- For the discussion section, I suggest simplifying the style and make more readable the text, without long sentences.
We have revised the sentence structure and made them shorter where applicable throughout the manuscript.
Reviewer 3 Report
The manuscript plants-1430250 evaluated the competitive characteristics of barley cultivars for suppressing weeds. The experimental work was well designed, and the authors obtained interesting data over three years. However, the presentation and description of the results is very ambiguous. The authors abuse a very subjective language and lack clarity in many of their statements. In this regard, it is important that each statement made about the results is statistically supported, but instead of reiterating whether or not a certain result was significant, the authors must indicate numerical information on the main results.
On the other hand, the quality of some figures and the table is poor, so these items must be redesigned. Furthermore, the legends of some figures have enough information for the sub-figures to be self-explanatory. Throughout the text it is also common to find undefined abbreviations at their first mention in the main text. In the attached document you can find my suggestions, but I recommend that you carefully review the manuscript to correct other minor typos.

Author Response
Reviewer 3
Thank you for your time and effort to read through our manuscript. Thank you also for your comments and suggestions on how we can improve the manuscript. Much appreciated
Line 19 Photosynthetically active light interception (PAR) has been deleted and used only PAR as it eas a repetition (line 19-20)
Line 32 significantly was deleted see line 36
Line 45 significantly was deleted see line 48
Line 46 potentially changed to may line 49
Line 58 typically deleted – see line 61
Line 104 NSW changed to New South Wales (NSW) line 107-108
Line 117 significantly deleted -see line 267
Line 120 significantly deleted -see line 270
Line 133 definition of DAE. This is provided in lines 191-192
Line 137 ‘was significantly different’ was deleted and replaced with ‘differed’ line 289
Line 146 PAR is defined in line 166
Other comments
The other comments and suggestions have been incorporated in the document during the revision and rewriting of the manuscript. It is possible some of the changes may have been missed, but we had all the intentions to respond to each one of them. Most acronyms were defined in the Materials and Methods, but during the transfer of the manuscript into the plant's template, this section came after discussion resulting in confusion. Now that the document has been changed the right way, we hope these definitions have been done in the correct order based on the first mention.
However, I would like to note that where significant and/or significantly words have been used, it is in reference to the P values 0.05, 0.01 0.001and it is not based on the scale of 1 to 10 or 100. Based on our previous publications and these words have been extensively used to describe the level of significance in that regard. In the manuscript, there are also other words used to describe differences, such as lower, higher and highest. Our suggestion in such cases is to refer to the tables or figures to read the difference. I hope this is why the tables and figures are provided for such comparisons. Otherwise, there will be a lot of numbers to present in the text.
Round 2
Reviewer 1 Report
This manuscript has been improved based on reviewer comments, thank you for your work.
Author Response
Thank you for your time and effort to read through our manuscript. We have revised according to your comments.
Please check revised version of manuscript in attachment. Much appreciated.

Reviewer 3 Report
The authors have slightly improved the presentation of their manuscript. However, the description of results remains extremely, markedly, strongly and relatively highly subjective. No relevant results are reported numerically in the abstract. The description is very ambiguous and limited to saying: "greater than", "less than" or "similar to", "strongly", "significantly", "not significantly different", etc. The same occurs in the results section and the authors did not attend to my observations in the previous version in relation to reporting the standard error values ​​and the ANOVA groups in Table 2. The authors argue that the use of ambiguous and subjective terms to describe their results is due to the fact that they are widely used in the scientific literature, but that these terms are common does not mean that they are the most scientifically appropriate. The description of results should be brief and precise, highlighting the main results numerically, and should not force the reader to consult tables and figures. Furthermore, some tables and figures and references are outside the journal's standards.

Author Response

(The authors gave the same response as above.)

Round 3
Reviewer 3 Report
Mwendwa et al. greatly improved the writing of their manuscript, however, it can still find some statements strongly, strikingly, alarmingly, extremely highly subjective. As they are terms widely used throughout the scientific literature, I am completely convinced that authors know what I mean. But to better understand me, rating their "subjective" statements from 0 to 10, each statement could receive a score of 0.6 to 9.9 if p> 0.5 is considered statistically significant.
L19: define ‘strong’ scientifically or use a numerical reference for your declaration
L23: in what proportion was photosynthesis increased? indicate numerically the observed result
L24: In what proportion does the accumulation of dry matter increase and in what percentage does it interfere with weeds?
L46: change ‘to herbicide resistance’ by ‘to resistant weeds’
L361: delete ‘significantly’ and indicate numerically by what percentage the weed suppression was improved
Tables and figures are elements that generally group the results of the investigation and undoubtedly all the descriptions highlighted on them are reflected. Although these elements are published together with the manuscript, they must be considered as an appendix to the text, that is, the text must be sufficiently understandable in such a way that it does not force the reader to consult tables and figures. For this reason, only the most relevant results of the research are described and discussed, and not each individual result. Conversely, the organization of tables and figures and their descriptions (title or footnotes) should be self-explanatory and not force the reader to consult the main text.
Author Response

(The authors gave the same response as above.)
